# Towards Practical Human Motion Prediction with LiDAR Point Clouds

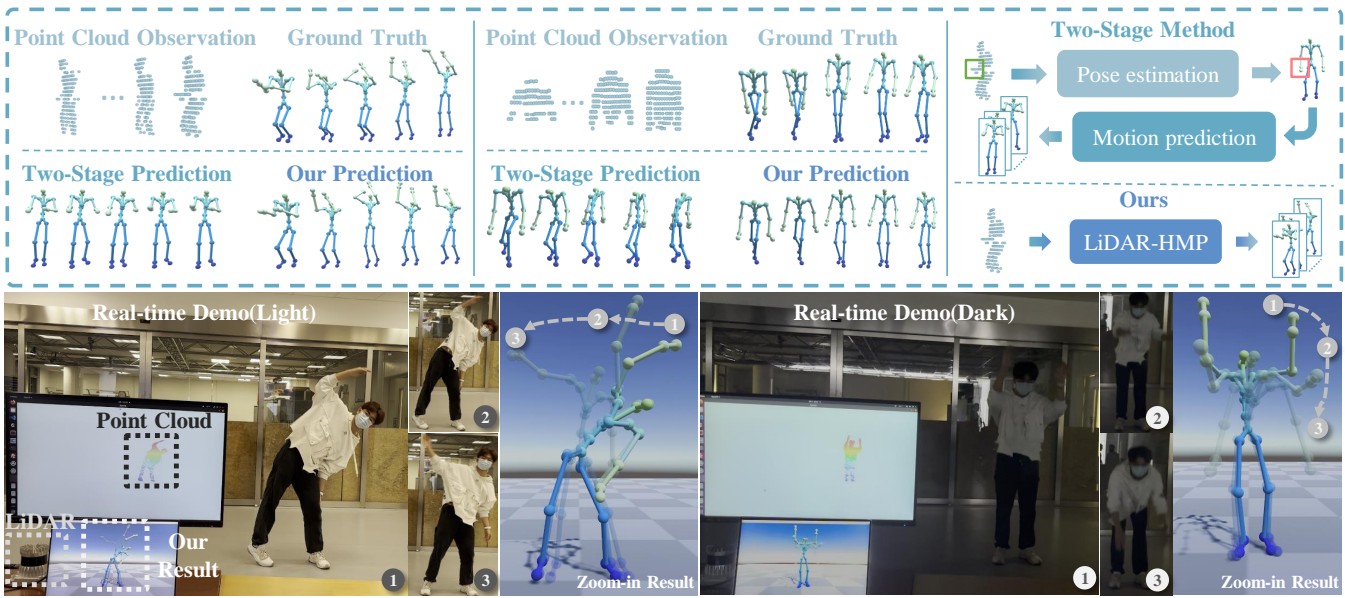

**Figure 1: Visualization of our motion prediction performance. The top figure demonstrates the comparison of our LiDAR-HMP and two-stage method on LIPD test set. The bottom figure highlights LiDAR-HMP's practicality in real-world deployment, unfettered by lighting conditions, where markers 1, 2, and 3 indicate the current moment and predicted poses for the future 0.4s and 1.0s, respectively. With online captured LiDAR point cloud, our method achieves real-time promising prediction results, which is significant for real-world applications.**

## ABSTRACT

Human motion prediction is crucial for human-centric multimedia understanding and interacting. Current methods typically rely on ground truth human poses as observed input, which is not practical for real-world scenarios where only raw visual sensor data is available. To implement these methods in practice, a pre-phrase of pose estimation is essential. However, such two-stage approaches often lead to performance degradation due to the accumulation of errors. Moreover, reducing raw visual data to sparse keypoint representations significantly diminishes the density of information, resulting in the loss of fine-grained features. In this paper, we propose *LiDAR-HMP*, the first single-LiDAR-based 3D human motion prediction approach, which receives the raw LiDAR point cloud as input and forecasts future 3D human poses directly. Building upon our novel structure-aware body feature descriptor, LiDAR-HMP adaptively maps the observed motion manifold to future poses and effectively models the spatial-temporal correlations of human motions for further refinement of prediction results. Extensive experiments show that our method achieves state-of-the-art performance on two public benchmarks and demonstrates remarkable robustness and efficacy in real-world deployments.

## CCS CONCEPTS

• **Computing methodologies** → **Temporal reasoning**; **Activity recognition and understanding**.

## KEYWORDS

Human motion prediction, Multimedia understanding, LiDAR point cloud

*ACM MM, 2024, Melbourne, Australia*
© 2024 Copyright held by the owner/author(s). Publication rights licensed to ACM.
ACM ISBN 978-x-xxxx-xxxx-x/YY/MM
https://doi.org/10.1145/nnnnnnn.nnnnnnn

## 1 INTRODUCTION

Bridging the gap between human actions and digital interactions, human motion prediction stands as a pivotal innovation in the ever-evolving field of multimedia. By accurately predicting future human movements through the analysis of past motion sequences, people are able to create multimedia content that is more intuitive,

responsive, and engaging. For instance, fitness apps can offer personalized guidance and feedback in real-time based on the predicted future movements of their users. Assistive robots can predict human intentions and proactively handover objects to users as they reach out, instead of waiting until they stop moving.

Current human motion prediction methods [1, 12, 28, 38, 44, 46, 47, 52] have achieved remarkable success by explicitly modeling the inherent body structural and dynamic features among sequential human skeletons. However, these approaches heavily rely on precise historical human poses, necessitating complex setups like multi-view cameras or densely deployed wearable IMUs, thus imposing substantial limitations on their practical application and deployment. In multimedia applications such as augmented reality(AR) environments, the system is expected to interpret human actions and gestures in real-time using their onboard sensors, with raw data often comprising images or point clouds from a single viewpoint. The dependency on the accuracy of past motion estimation techniques [17, 22, 36, 45, 50, 55] leads to **cumulative errors** as shown in the upper-right corner of Fig. 1 and the **loss of fine-grained features** present in the raw visual data. To advance practical human motion prediction, directly taking the raw visual data as input is a more viable strategy. Some studies [39, 51] have investigated video inputs to improve applicability, the absence of precise depth information and sensitivity to lighting conditions frequently results in ambiguous and inconsistent predictions in unconstrained environments.

In recent years, LiDAR's long-range depth-sensing and light-insensitive capabilities have established it as an indispensable sensor in the realm of autonomous driving [4, 23, 33, 43] and various human-centric applications [17, 21, 36, 37, 53], such as 3D pose estimation, human motion capture, gait recognition, action recognition, etc. Studies in these areas have underscored the superiority of LiDAR over traditional cameras, highlighting its robustness and effectiveness due to its ability to provide precise 3D geometry and dynamic motion information about humans in free environment. Crucially, LiDAR is not affected by lighting conditions, presenting a significant advantage for real-world deployment. Based on this, we introduce ***LiDAR-HMP***, the first LiDAR-based human motion prediction solution, which directly predicts future human poses from raw LiDAR point clouds and is more practical for real-world multimedia applications.

Our approach is founded on three carefully crafted modules that effectively harness the valuable information contained within the raw point cloud input, enabling precise predictions of future human motion as shown in Fig. 1. Firstly, the ***Structure-aware Body Feature Descriptor*** delves into the implicit semantics of human body structure entailed by the point cloud observation, amalgamating part-wise with global body features. This ensures a rich, detailed representation of human structure essential for accurate motion forecasting. Following this, our ***Adaptive Motion Latent Mapping*** module employs a set of learnable queries, each mapping to a future motion frame. These queries dynamically extract critical information from the observed point cloud frames, laying the foundation for precise motion prediction. Finally, the ***Spatial-Temporal Correlations Refinement*** module embeds coarse motion predictions and part-wise body features into a high-dimensional space. Leveraging

spatial-temporal correlations among the predicted poses, it meticulously refines the predicted keypoints, enhancing the prediction accuracy. Together, these modules embody our design philosophy of capturing both the spatial structure and temporal dynamics of human motion, setting a new benchmark for LiDAR-based human motion prediction in real-world robotic applications.

Notably, acknowledging the inherent unpredictability and potential for sudden changes in human movement, we extend our model to support diverse motion predictions [9, 24, 27, 32, 49], thus broadening its applicability and enhancing predictive reliability and robustness. Our experimental results and visual analysis both indicate that diverse predictions not only enhance the robustness and reliability of forecasts but also offer invaluable insights for applications requiring an in-depth understanding of human dynamics. For instance, in interactive systems and autonomous navigation, the ability to predict a range of potential motions can significantly improve the decision-making process.

We conduct experiments on both short-term and long-term human motion prediction on two public LiDAR-based human motion datasets, including LiDARHuman26M [17] and LIPD [53]. Our method significantly outperforms existing methods in terms of mean per joint position error (MPJPE) by a large margin, e.g. average 17.42$mm$ and 7.86$mm$ for short-term prediction, and average 11.62$mm$ and 9.12$mm$ for long-term prediction, respectively. We also evaluate the robustness and generalization ability of our method under long-range distance, occluded and noisy cases. In particular, we have deployed our method in real scenarios and achieved real-time human motion predictions, providing a feasible interface for further robotic applications. Our contributions can be summarized as follows:

- We propose the first LiDAR-based method for practical 3D human motion prediction by fully utilizing the fine-grained motion details in raw LiDAR point clouds, which closely aligns with real-world application scenarios.
- We present a structure-aware body feature descriptor that decouples the holistic human point cloud into distinct body parts based on the semantics of anatomical structure, enabling the capture of fine-grained dynamic motion details.
- We map the past motion manifold to the future motion manifold adaptively, and fully leverage the spatial-temporal correlation of the motions to further refine the predicted results.
- Our method achieves state-of-the-art performance on two public LiDAR-based human motion datasets.

## 2 RELATED WORK

### 2.1 Human Motion Prediction

Existing human motion prediction methods focus on forecasting future human motions based on past ground truth observations captured through multi-view cameras or dense IMUs, with specialized network structures to address the spatial and temporal dependencies in structured skeleton sequences. Several methods [11, 13, 16, 25, 31, 47] have explored modeling temporal dependencies in human motion prediction using RNN and TCN structures, but often overlook spatial relationships. Recent advancements [3, 8, 19, 42, 52] have seen GCNs achieving state-of-the-art results by learning spatial dependencies among joints with learnable weights. Building

on the GCN framework, DMGNN [19] and MSR-GCN [8] introduced multi-scale body graphs to capture both local and global spatial features. PGBIG [28] extended this approach by incorporating temporal graph convolutions for extracting spatial-temporal features. SPGSN [18] introduced a graph scattering network to enhance the modeling of temporal dependencies through multiple graph spectrum bands. DMAG [12] uses frequency decomposition and feature aggregation respectively to encode the information. Beyond GCN-based approaches, transformer architectures [1, 2, 30] also have been utilized to model pair-wise spatial-temporal dependencies, indicating a broad interest in capturing complex spatial-temporal relationships in human motion prediction. AuxFormer [46] introduced a model learning framework with an auxiliary task that can recover corrupted coordinates depending on the rest coordinates. [44] synthesizes the modeling ability of the self-attention mechanism and the effectiveness of graph neural networks. However, these methods are primarily designed for inputs of ground truth observed skeletons and showed performance degradation with inputs of estimated skeletons from 3D pose estimation methods [14, 17, 22, 26, 34, 36, 45, 50, 55] due to accumulative errors, restricting their real-world applicability. Although some studies [39, 51] have explored the use of video inputs to enhance practical applicability, the absence of precise depth information and sensitivity to lighting changes frequently result in ambiguous and inconsistent predictions in uncontrolled settings.

## 2.2 LiDAR-based Human-centric Applications

Due to the accurate depth sensing and light-insensitive ability in long-range scenes, LiDAR has emerged as a pivotal perception sensor for robots and autonomous driving [4, 15, 23, 33, 43, 54]. In recent years, numerous human-centric applications[5–7, 17, 21, 36, 37, 48, 53], such as pose estimation, motion capture, action recognition, gait recognition, scene reconstruction, etc., have adopted LiDAR to expand usage scenarios and improve performance of solutions by utilizing accurate geometric characteristics of LiDAR point clouds. Especially, recent works [17, 36, 53] have already underscored the efficacy of single-LiDAR systems for human motion capture. Given LiDAR's success in large-scale human-related applications, we make the first attempt to leverage LiDAR for practical 3D human motion predictions by fully exploiting the dynamic spatial-temporal correlations among observed sequential LiDAR point clouds.

## 3 METHODOLOGY

In this section, we introduce the details of our single-LiDAR-based 3D human motion prediction method LiDAR-HMP. After defining the problem in Sec.3.1, we introduce our components sequentially. In Sec.3.2, we introduce Structure-aware Body Feature Descriptor to extract comprehensive human features. Subsequently, in Sec.3.3, we employ Adaptive Motion Latent Mapping to facilitate dynamic interactions crucial for accurate motion prediction. Upon obtaining the initial predicted 3D skeleton joints, the Spatial-temporal Correlations Refinement module (Sec.3.4) further refines the correlations between different body joints. Finally, Sec.3.5 outlines the decoding and supervision of the network, while Sec.3.6 demonstrates the potential of our methods across various settings.

### 3.1 Problem Definition

3D human motion prediction involves forecasting 3D human motion poses for the next few frames based on a few observed frames. Let $\mathbf{P}_{1:T_o} = \left\{ P_1, P_2, \ldots, P_{T_o} \right\}$ represent the historical point cloud observation of length $T_o$, where $P_t$ denotes the point cloud frame at time $t$. Additionally, denote $\mathbf{J}_{T_o+1:T_o+T_p} = \left\{ J_{T_o+1}, J_{T_o+2}, \ldots, J_{T_o+T_p} \right\}$ as the predicted 3D pose sequence of length $T_p$. Here $J_t \in \mathbb{R}^{24 \times 3}$ signifies the future human pose of time $t$, represented by 24 3D keypoint coordinates. Our approach aims to predict the $T_p$ frames of future 3D human motion poses from the $T_o$ observed frames of human LiDAR point clouds.

### 3.2 Structure-aware Body Feature Descriptor

Contrary to the inherently structured skeleton-based human motion representations, the sparse and disordered nature of LiDAR point clouds pose substantial challenges for efficient feature extraction. Existing methods in human-focused research [36, 53] commonly adopt PointNet [35] to extract bodily features from point clouds, using max pooling to distill key features from the LiDAR data. While efficient and widely applied in existing LiDAR-based human analytics, such an approach overlooks the rich geometric and structural details in the 3D point cloud data, which are essential for deriving expressive features that accurately represent human body structure. To overcome these limitations, we introduce the Structure-aware Body Feature Descriptor. Our approach combines the holistic global feature of the entire body with detailed local features that capture the semantics of anatomical body parts, including the thigh, calf, arm, and more. This decomposition enables a nuanced and comprehensive representation of human motion, facilitating a deeper understanding of complex movements.

Specifically, given a frame of point cloud observation of the past human motion $P_i = \{p_1, ..., p_N\} \in \mathbf{P}_{1:T_o}$, we segment the human body point cloud into $K$ semantically meaningful anatomical parts via a pre-trained human part segmentation model $\Phi(\cdot)$. *Details of the pre-trained human parsing model are illustrated in the supplementary materials.* We retrieve the point-wise features $F_i = \{f_1, f_2, \ldots, f_N\}$ from the PointNet encoder before the global max-pooling. These point-wise features are then scattered into $K$ bins corresponding to different body part segments. Within each bin, we perform max-pooling across the points to obtain part-wise feature $H_{part} \in \mathbb{R}^{T_o \times K \times d_1}$:

$$H_{part} = \left\{ H_{part}^k = \text{MaxPool}(\{f_i | \Phi(p_i) == k\} \big| k = 1, 2, \ldots, K \right\}. \quad (1)$$

We then integrate the global body feature $H_{glo}$ yielded by the Point-Net encoder with the part-wise feature to obtain the structure-aware body feature descriptor $H$:

$$H = \left[ H_{glo} \oplus H_{part} \right] \in \mathbb{R}^{T_o \times (K+1) \times d_1}, \quad (2)$$

where $\oplus$ denotes concatenation. The feature descriptor is further enhanced with a spatial-temporal transformer layer, which models the non-linear dependencies among the body parts across different frames, further refining the descriptor's ability to capture complex bodily dynamics.

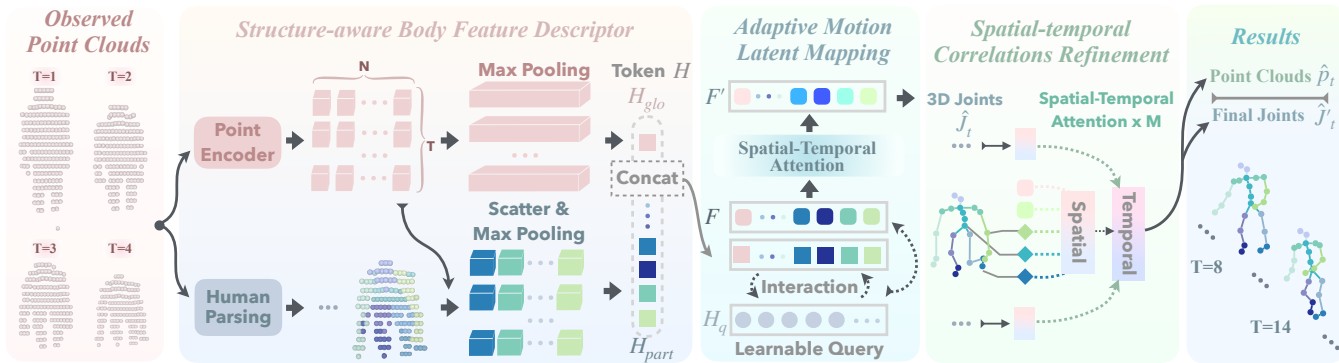

**Figure 2: The pipeline of our LiDAR-HMP. First, we obtain the structure-aware body feature descriptor from the observed LiDAR point cloud frames. Then, we adaptively predict the human motion with learnable queries for initial predictions and explicitly model the spatial-temporal correlations among them to refine the predicted motions. Finally, we decode the joint-wise results and point-wise results for auxiliary supervision.**

### 3.3 Adaptive Motion Latent Mapping

Building upon the expressive feature descriptor of the observed LiDAR point cloud frames, we advance towards predicting the future 3D human motion from historical motion observations. Specifically, we initialize a set of learnable motion queries $H_q \in \mathbb{R}^{(T_o+T_p) \times (K+1) \times d_1}$ following a normal distribution $\mathcal{N}(0, 0.02)$, designated for each of the observed and predicted frames. These queries then selectively assimilate information from past frame descriptors within a transformer decoder layer to translate past frame features into a comprehensive feature set that encapsulates both past observations and future predictions, thereby laying the foundation for accurate human motion forecasting:

$$F = \text{MHCA}\left(Q: W_q H_q, \ K: W_k H, V: W_v H\right), \quad (3)$$

where $\text{MHCA}(\cdot \text{ query}, \cdot \text{ key}, \cdot \text{ value})$ represents the multi-head cross-attention layer [41] and $W_q, W_k, W_v$ are the linear projection for query, key and value. To further reinforce the spatial-temporal consistency within the updated body features, we apply spatial and temporal transformer layers, progressively refining the predicted motion features $F$ into $F'$. Finally, we regress the coarse 3D joints $\hat{J}_t \in \mathbb{R}^{24 \times 3}$ through the refined motion features $F'$. The loss function can be formulated as below:

$$\mathcal{L}_{\text{initial}} = \sum_{t=1}^{T_o+T_p} \left\| \hat{J}_t - J_t \right\|^2. \quad (4)$$

### 3.4 Spatial-temporal Correlations Refinement

Leveraging the aforementioned modules, we derive the initial predicted 3D skeletal joints. However, given the structured nature of human joint motion, it is crucial to explicitly model the spatial-temporal correlations among various human joints for accurate motion prediction. To address this, we employ a spatial-temporal transformer to integrate joint features, local semantic features, and global geometric features. This facilitates dynamic refinement of these correlations, effectively modeling the spatio-temporal contextual information. Specifically, we first embed the coarse 3D pose prediction $\hat{\mathbf{J}} = \{\hat{J}_1, \hat{J}_2, \ldots, \hat{J}_T\}$ into joint-wise features using an

Multilayer Perceptron(MLP) layer:

$$E_{\text{joint}} = \text{MLP}(\hat{\mathbf{J}}) \in \mathbb{R}^{T \times 24 \times d_2}, \quad (5)$$

where $T = T_o + T_p$. Similarly, we transform the refined motion features $F'$ to the same dimension:

$$E_F = \text{MLP}(F') \in \mathbb{R}^{T \times (K+1) \times d_2}. \quad (6)$$

These are concatenated to form tokens $TK = \left[E_{\text{joint}} \oplus E_F\right]$ for the transformer layers. To explore implicit information within these tokens, we deploy a spatial transformer (STFormer) $F_{STF}$ and a temporal transformer (TTFormer) $F_{TTF}$ to capture the dependencies within spatial and temporal dimensions, respectively. The STFormer aims to model the spatial dependencies among all tokens within a frame. We slice the token embedding $TK$ at the spatial dimension and encode the $i^{\text{th}}$ token $TK^i$ with a learnable spatial positional encoding to signify each token's relative spatial location before processing by STFormer. Utilizing the self-attention mechanism, STFormer models the dependencies of all tokens per frame, yielding spatially consecutive token-wise features as below:

$$H_s = \left\{ F_{STF}\left(TK^{1:(K+25)}\right)_t \Big|_{t=1}^{T} \right\} \in \mathbb{R}^{T \times (K+25) \times d_2}. \quad (7)$$

Conversely, TTFormer is tasked with discerning the temporal correlations to ensure temporal consistency and motion accuracy. It regards each feature across the sequence as tokens, generating features $TK_t \in \mathbb{R}^{(K+25) \times d_2}$ with a total of $T$ tokens, where $TK_t$ represents the $t^{\text{th}}$ token of the token embedding $TK$ sliced in the temporal dimension. Additionally, we append a learnable temporal positional encoding to $TK^i$ to mark each token's temporal position in the sequence. The TTFormer then encodes the temporally modeled motion features as:

$$H_t = \left\{ F_{TTF}\left(TK_{1:T}\right)^i \Big|_{i=1}^{(K+25)} \right\} \in \mathbb{R}^{T \times (K+25) \times d_2}. \quad (8)$$

We adopt STFormer and TTFormer layers alternatively, which enables a thorough modeling of spatial-temporal correlations. STFormer layers effectively capture the complex spatial relationships present in the data, whereas TTFormer layers adeptly model the temporal dynamics of human motion. By focusing on both spatial and temporal aspects, essentially the "where" and the "when" of motion, our

**Table 1: Short-term motion prediction results of MPJPE(mm) on LIPD [53] and LiDARHuman26M [17] datasets. "AVG" means the average MPJPE results of 100ms, 200ms, 300ms and 400ms.**

| Methods | Publication Year | LIPD [53] | | | | | LiDARHuman26M [17] | | | | |
|---|---|---|---|---|---|---|---|---|---|---|---|
| | | 100ms↓ | 200ms↓ | 300ms↓ | 400ms↓ | **AVG↓** | 100ms↓ | 200ms↓ | 300ms↓ | 400ms↓ | **AVG↓** |
| LiDARCap [17]+SPGCN [28] | 2022 / 2022 | 82.63 | 100.98 | 116.71 | 128.67 | 107.25 | 83.69 | 94.68 | 105.09 | 113.97 | 99.35 |
| LiDARCap [17]+Eqmotion [47] | 2022 / 2023 | 81.27 | 97.26 | 111.28 | 122.79 | 103.15 | 83.01 | 91.66 | 101.15 | 109.91 | 96.43 |
| LiDARCap [17]+AuxFormer [46] | 2022 / 2023 | 82.09 | 99.19 | 113.80 | 124.89 | 104.99 | 83.77 | 94.90 | 105.36 | 113.96 | 99.50 |
| LIP [53]+SPGCN [28] | 2023 / 2022 | 77.75 | 95.67 | 111.40 | 123.68 | 102.12 | 81.20 | 91.43 | 101.22 | 110.07 | 95.98 |
| LIP [53]+Eqmotion [47] | 2023 / 2023 | 77.34 | 93.78 | 108.19 | 120.18 | 99.87 | 81.14 | 89.76 | 98.80 | 107.49 | 94.30 |
| LIP [53]+AuxFormer [46] | 2023 / 2023 | 77.81 | 95.00 | 109.85 | 121.18 | 100.96 | 81.31 | 91.70 | 101.77 | 110.25 | 96.26 |
| LiveHPS [36]+SPGCN [28] | 2024 / 2022 | 70.40 | 87.83 | 103.62 | 116.21 | 94.51 | 76.29 | 84.87 | 93.89 | 101.73 | 89.20 |
| LiveHPS [36]+Eqmotion [47] | 2024 / 2023 | 71.60 | 87.75 | 102.67 | 115.40 | 94.22 | 76.97 | 84.17 | 92.50 | 101.14 | 88.69 |
| LiveHPS [36]+AuxFormer [46] | 2024 / 2023 | 70.89 | 88.31 | 103.69 | 115.55 | 94.61 | 77.13 | 86.65 | 96.62 | 104.40 | 91.20 |
| MDM [40] | 2023 | 77.38 | 89.28 | 100.73 | 110.66 | 94.51 | 73.12 | 78.21 | 85.65 | 92.68 | 82.41 |
| Ours | | **60.05** | **70.84** | **82.65** | **93.70** | **76.80** | **67.09** | **69.88** | **76.57** | **84.64** | **74.55** |

approach achieves a comprehensive analysis and synthesis, leading to more accurate and coherent motion predictions.

## 3.5 Motion Prediction Head

In the network's final phase, we employ dual heads to decode human skeletons and point clouds concurrently, utilizing Multilayer Perceptrons (MLPs). The joint-wise regression head utilizes three MLP layers to decode the final joints $\hat{J}'_t \in \mathbb{R}^{24 \times 3}$. To supervise the training of this head, we apply an $L_2$ loss:

$$\mathcal{L}_{\text{final}} = \sum_{t=1}^{T_o+T_p} \left\| \hat{J}'_t - J_t \right\|^2. \tag{9}$$

For the point-wise regression head, three MLP layers are employed to decode each local-semantic feature into points corresponding to each human body part. With $K$ predefined body parts, we achieve the final point clouds representation $\hat{P}'_t \in \mathbb{R}^{K \times 32 \times 3}$. We compare the point cloud obtained from the network outputs $\hat{P}_t = \left\{ \hat{p} \in \mathbb{R}^3 \right\}$, $\left| \hat{P}_t \right| = N$, with $\hat{P}_t$ with the ground truth point cloud $P_t = \left\{ p \in \mathbb{R}^3 \right\}$, $|P_t| = M$ by Chamfer Distance as follow:

$$\mathcal{L}_{\text{CD},t} = \frac{1}{N} \sum_{\hat{p} \in \hat{P}_t} \min_{p \in P_t} \|\hat{p} - p\|_2^2 + \frac{1}{M} \sum_{p \in P_t} \min_{\hat{p} \in \hat{P}_t} \|\hat{p} - p\|_2^2. \tag{10}$$

The overall loss is:

$$\mathcal{L} = \mathcal{L}_{\text{initial}} + \mathcal{L}_{\text{final}} + \mathcal{L}_{\text{CD},t}. \tag{11}$$

## 3.6 Diverse Motion Prediction Extension

Given the highly subjective nature of human behaviour, motion prediction requires multiple potential future predictions. We extend our approach to diverse human motion prediction setting. Following the above framework, we implement $K = 4$ multiple learnable motion queries in our adaptive motion intention prediction module. For each probable motion feature, they share all the network weight during the training process. We can regard each learnable query as a different motion mode that can map the past motion into

multiple future motion latent space with various motion patterns. The "winner-take-all" (WTA) training strategy [10, 20] is employed, which only optimizes the best prediction with minimal average prediction error to the ground truth human motion.

## 4 EXPERIMENTS

In this section, we begin by introducing the datasets (Sec.4.1), implementation details (Sec.4.2). Subsequently, we conduct a qualitative and quantitative comparison between our method and current state-of-the-art (SOTA) methods on two publicly available LiDAR-based human motion datasets (Sec.4.3), showcasing the superiority and generalization capabilities of our approach. Additionally, we conduct ablation studies to illustrate the superiority of our network design (Sec.4.4). Moreover, we analyze the robustness and generalization ability of our method (Sec.4.5). Finally, we present experimental results on diverse human motion predictions in Sec.4.6, highlighting its importance and utility in real-world scenarios, particularly in long-term prediction, where human behavior is highly unpredictable and emergent.

## 4.1 Datasets

LIPD [53] is a comprehensive dataset designed for LiDAR-related motion capture and focuses on a variety of challenging motions. It includes data from 15 performers executing around 30 motion types, amounting to 62,341 frames of LiDAR point clouds. LiDARHuman26M [17] includes contributions from 13 volunteers, consisting of 11 males and 2 females. Each volunteer participated in sessions ranging from 15 to 30 minutes, performing 20 types of daily motions such as walking, swimming, running, phoning, and bowing. The dataset contains 184,048 frames with a total of 26,414,383 points.

## 4.2 Implementation Details

We build our network on PyTorch 2.1.0 and CUDA 12.1, trained over 100 epochs with batch size of 128, using an initial learning rate of $10^{-4}$. We set the hyper parameters of our model as {$K$ =

**Table 2: Long-term motion prediction results of MPJPE(mm) on LIPD[53] and LiDARHuman26M [17] datasets. "AVG" means the average MPJPE results of 600ms, 800ms and 1000ms.**

| Methods | Publication Year | LIPD [53] | | | | LiDARHuman26M [17] | | | |
|---|---|---|---|---|---|---|---|---|---|
| | | 600ms↓ | 800ms↓ | 1000ms↓ | **AVG↓** | 600ms↓ | 800ms↓ | 1000ms↓ | **AVG↓** |
| LiDARCap [17]+SPGCN [28] | 2022 / 2022 | 142.82 | 153.70 | 159.64 | 152.05 | 125.42 | 134.03 | 137.89 | 132.44 |
| LiDARCap [17]+Eqmotion [47] | 2022 / 2023 | 138.83 | 147.95 | 154.44 | 146.91 | 120.93 | 126.85 | 131.36 | 126.38 |
| LiDARCap [17]+AuxFormer [46] | 2022 / 2023 | 137.10 | 147.15 | 153.54 | 145.93 | 122.86 | 129.72 | 134.05 | 128.87 |
| LIP [53]+SPGCN [28] | 2023 / 2022 | 138.08 | 149.46 | 155.76 | 147.76 | 120.79 | 129.18 | 133.27 | 127.75 |
| LIP [53]+Eqmotion [47] | 2023 / 2023 | 136.03 | 146.18 | 153.04 | 145.08 | 118.84 | 125.25 | 129.81 | 126.63 |
| LIP [53]+AuxFormer [46] | 2023 / 2023 | 133.79 | 144.11 | 150.79 | 142.89 | 118.73 | 125.60 | 130.39 | 124.91 |
| LiveHPS [36]+SPGCN [28] | 2024 / 2022 | 130.66 | 142.65 | 150.07 | 141.13 | 111.79 | 119.97 | 124.71 | 118.82 |
| LiveHPS [36]+Eqmotion [47] | 2024 / 2023 | 132.21 | 143.41 | 151.30 | 142.31 | 113.37 | 120.30 | 125.74 | 119.80 |
| LiveHPS [36]+AuxFormer [46] | 2024 / 2023 | 127.03 | 137.92 | 145.74 | 136.90 | 112.40 | 119.39 | 124.22 | 118.67 |
| MDM [40] | 2023 | 139.58 | 162.51 | 173.69 | 158.59 | 124.05 | 149.01 | 158.39 | 143.82 |
| Ours | | **113.24** | **126.49** | **136.10** | **125.28** | **100.85** | **110.81** | **117.00** | **109.55** |

**Table 3: Ablation studies for network design on LIPD dataset, where "SBFD" denotes our structure-aware body feature descriptor module, and "STCR" represents the spatial-temporal correlations refinement module.**

| Network Module | | | | short-term | | | | | long-term | | | |
|---|---|---|---|---|---|---|---|---|---|---|---|---|
| Baseline | SBFD | STCR | point prediction | 100ms | 200ms | 300ms | 400ms | AVG | 600ms | 800ms | 1000ms | AVG |
| ✓ | | | | 64.49 | 75.44 | 87.48 | 98.71 | 81.53 | 116.60 | 128.96 | 138.08 | 127.88 |
| ✓ | ✓ | | | 61.91 | 73.11 | 84.64 | 95.87 | 78.88 | 114.44 | 126.91 | 136.17 | 125.84 |
| ✓ | ✓ | ✓ | | 60.46 | 71.30 | 83.08 | 94.21 | 77.26 | 113.10 | 125.78 | 135.33 | 124.74 |
| ✓ | ✓ | ✓ | ✓ | **60.05** | **70.84** | **82.65** | **93.70** | **76.80** | **112.84** | **125.51** | **135.11** | **124.49** |

$9, d_1 = 1024, d_2 = 512$} across all experiments. The process was run on a server equipped with two Intel(R) Xeon(R) E5-2678 CPUs and 4 NVIDIA RTX3090 GPUs. We resample each frame of input point clouds $P_i$ to a fixed $N = 256$ points by the farthest point sample algorithm, then we subtract the center coordinates of point clouds to normalize the input data. Following skeleton-based human motion prediction methods [18, 28], our model takes 0.4s continuous point clouds as input(4 frames for LiDAR with 10fps), and we predict the future 0.4s(4 frames) and 1.0s(10 frames) human motion for short-term and long-term prediction, respectively. For compared methods, we keep the same training strategy as in the original paper. For training data, we take training set of LiDAR-related human motion dataset LIPD [53], LiDARHuman26M [17], and synthetic dataset of a subset of AMASS [29], including ACCAD, BMLMovi, CMU following LIP [53]. We leverage Mean Per Joint Position Error (MPJPE) in millimeters as our evaluation metric, which is a common metric used for evaluating the accuracy of predicted 3D human poses against ground truth data.

## 4.3 Results

We evaluate LiDAR-HMP against several state-of-the-art methods [17, 28, 36, 40, 46, 47, 53] on two public datasets (LIPD [53] and LiDARHuman26M [17]) to demonstrate its superiority in 3D human motion prediction task. The results for short-term and long-term human motion prediction are shown in Tab. 1 and Tab. 2, respectively.

*(1) Comparison with two-stage methods.* Conventional methods for human motion prediction typically rely on historical ground truth skeletons as input. In contrast, our method utilizes historical LiDAR frames alone. We integrate results from state-of-the-art LiDAR-based 3D human pose estimation methods into these skeleton-based approaches. Due to accumulative error and the loss of fine-grained features, two-stage methods often show degraded performance in real-world scenarios that depend solely on raw sensor data. In contrast, our approach demonstrates substantial improvements, achieving enhancements of average 17.42mm and 7.86mm in short-term human motion prediction, and average

**Table 4: Ablation of structure-aware body feature descriptor on LIPD dataset.**

| global | local | 100 | 200 | 300 | 400 | 600 | 800 | 1000 |
|:---:|:---:|:---:|:---:|:---:|:---:|:---:|:---:|:---:|
| ✓ | | 63.40 | 74.65 | 86.88 | 98.08 | 116.28 | 128.74 | 137.78 |
| | ✓ | 60.67 | 71.42 | 82.93 | 93.90 | 114.26 | 127.05 | 136.57 |
| ✓ | ✓ | **60.05** | **70.84** | **82.65** | **93.70** | **112.84** | **125.51** | **135.11** |

**Table 5: Ablation of adaptive motion latent mapping module on LIPD dataset.**

| | 100 | 200 | 300 | 400 | 600 | 800 | 1000 |
|:---|:---:|:---:|:---:|:---:|:---:|:---:|:---:|
| static padding | 60.95 | 71.47 | 83.11 | 94.27 | 113.21 | 126.33 | 136.39 |
| linear mapping | 60.94 | 71.93 | 83.62 | 94.70 | 113.58 | 126.29 | 136.49 |
| ours | **60.05** | **70.84** | **82.65** | **93.70** | **112.84** | **125.51** | **135.11** |

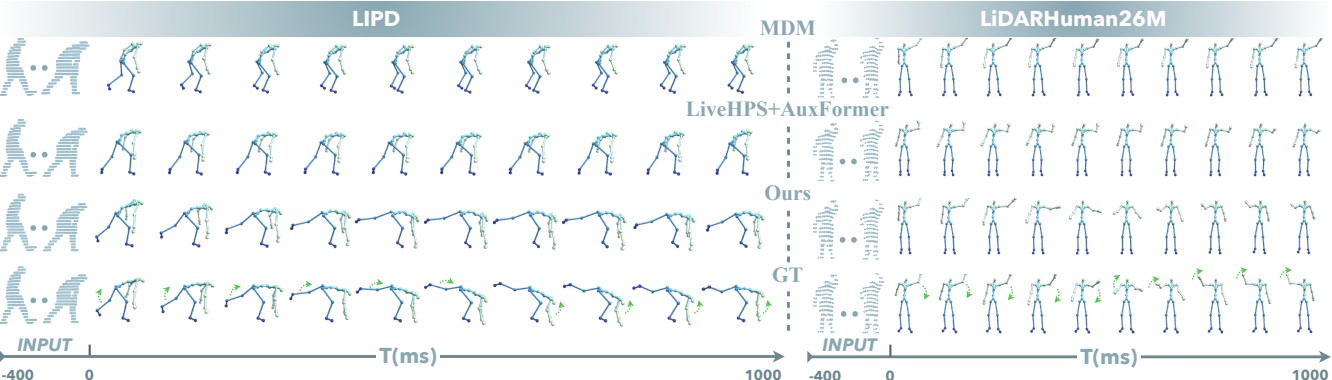

**Figure 3: Qualitative comparisons of long-term predictions on LIPD and LiDARHuman26M dataset. "GT" denotes the future ground truth skeletons. The green arrow denotes the motion trace.**

11.62mm and 9.12mm in long-term prediction on the LIPD and Li-DARHuman26M datasets, respectively. This superior performance stems from our method's capability to fully capture the dynamic spatial-temporal motion features from raw LiDAR point clouds. Our approach not only bypasses the inaccuracies introduced by intermediate skeletons from 3D pose estimation methods, but also retains the fine-grained, point-level features from the raw LiDAR data.

*(2) Comparison with diffusion-based method.* In addition, we also adapt the Motion Diffusion Model (MDM) [40] methodology for LiDAR-based human motion prediction by substituting its text encoder with a PointNet [35] encoder for another comparison. From Tab. 1 and Tab. 2, we can observe that diffusion-based methods obtain limited performance. This is mainly because the diffusion-based method relies on gradually denoising a signal towards generating a coherent output, may struggle with the variability and noise inherent in point cloud data. The nuances of human motion, particularly subtle movements are lost or inadequately captured during the diffusion process.

Furthermore, visual comparisons in Fig. 3 highlight our method's superiority in predicting long-term complex motions. Such as predicting the motion of bending down and getting up in LIPD and the motion of the lowering left hand and raising the right hand in LiDARHuman26M, MDM and two-stage method can only predict short-term motions or remain consistent with historical motions. By effectively modeling dynamic spatial-temporal correlations among structure-aware motion features from raw visual input, our approach offers enhanced accuracy and applicability in real-world scenarios of 3D human motion prediction.

## 4.4 Ablation on Network and Module Design

*(1) Ablation on overall network design.* We assess the efficacy of our proposed modules through comprehensive ablation studies on LIPD [53] dataset, as detailed in Tab. 3. Notably, our baseline models focus solely on global-geometric feature extraction and incorporate the adaptive motion latent mapping module. *Incorporating fine-grained local-semantic features* leads to average 2.65mm, 2.04mm improvements on short-term and long-term prediction, respectively. The addition of our *spatial-temporal correlations refinement module*, which explicitly models the spatial-temporal relationships among distinct body joints and structure-aware motion features, further refines our 3D human motion predictions. Introducing an *auxiliary points regression head* also contributes to performance improvements by providing additional supervision, thereby enhancing the network's predictive capabilities.

*(2) Structure-aware body feature descriptor.* The analysis presented in Tab. 4 underscores the importance of fine-grained local-semantic feature modeling, especially for capturing nuanced body movements. Our approach effectively leverages both holistic global-geometric motion features and detailed local-semantic information, delivering more robust and precise human motion predictions.

*(3) Adaptive motion latent mapping.* To validate the effectiveness of our adaptive motion latent mapping module, we conducted experiments by replacing the module's learnable motion query with a linear mapping in the temporal dimensions using an MLP. Additionally, we compare our method against a "static padding" approach, where future motions are extrapolated by replicating the last frame of observed motion at the input stage. As shown in Tab. 5, our approach achieves superior performance. This is attributed to our

**Table 6: MinMPJPE(mm) results of diverse motion prediction extension. "AVG" means the average MPJPE(mm).**

| motions | LIPD [53] | | | | | | LiDARHuman26M [17] | | | | | |
|---|---|---|---|---|---|---|---|---|---|---|---|---|
| | 200ms↓ | 400ms↓ | 600ms↓ | 800ms↓ | 1000ms↓ | AVG↓ | 200ms↓ | 400ms↓ | 600ms↓ | 800ms↓ | 1000ms↓ | AVG↓ |
| 1 | 70.84 | 93.70 | 113.24 | 126.49 | 136.10 | 108.07 | 69.88 | 84.64 | 102.16 | 112.51 | 118.40 | 97.51 |
| 4 | 68.61 | 87.49 | 98.25 | 106.03 | 117.68 | 95.61 | 64.57 | 78.09 | 91.16 | 98.25 | 107.21 | 87.85 |

**Table 7: Ablation on LIPD with simulated occlusion.**

| occlusion ratio | 100 | 200 | 300 | 400 | 600 | 800 | 1000 |
|---|---|---|---|---|---|---|---|
| 0% | 60.05 | 70.84 | 82.65 | 93.70 | 112.84 | 125.51 | 135.11 |
| 20% | 61.60 | 72.60 | 84.48 | 95.68 | 114.66 | 127.31 | 136.72 |
| 40% | 62.96 | 74.22 | 86.34 | 97.70 | 116.16 | 128.62 | 137.88 |
| 80% | 66.57 | 78.29 | 90.75 | 102.20 | 120.28 | 132.59 | 141.40 |

**Table 8: Ablation on LIPD with simulated noise.**

| noise ratio | 100 | 200 | 300 | 400 | 600 | 800 | 1000 |
|---|---|---|---|---|---|---|---|
| 0% | 60.05 | 70.84 | 82.65 | 93.70 | 112.84 | 125.51 | 135.11 |
| 20% | 61.49 | 72.52 | 84.55 | 95.81 | 113.97 | 126.63 | 136.18 |
| 40% | 62.29 | 73.29 | 85.24 | 96.42 | 114.43 | 127.27 | 136.96 |
| 80% | 63.02 | 74.02 | 85.93 | 97.13 | 115.47 | 128.21 | 137.55 |

learnable motion query's ability to selectively assimilate information from past motion features and more accurately project it into the future motion latent space.

### 4.5 Generalization Capability Test

*(1) Distance.* To assess the generalization capability of our LiDAR-HMP across point clouds of varying sparsity and distances, we evaluate our method on the LIPD testing sets. Performance of the short-term motion prediction at 0.4s and long-term motion prediction at 1.0s in different distances is illustrated in Figure 4. Despite the decreasing density of LiDAR human point clouds with increasing distance results in sparser representations, our method still maintains stable performance up to a long-range distance of 17 meters. This stability is crucial for large-scale applications such as autonomous driving and robot obstacle avoidance, where motion prediction at long ranges is essential. At about 20 meters, the point cloud may reduce to roughly 30 points, capturing only the basic human outline, which poses challenges for capturing detailed movements. Nonetheless, our method consistently delivers the best performance, even under these extreme conditions.

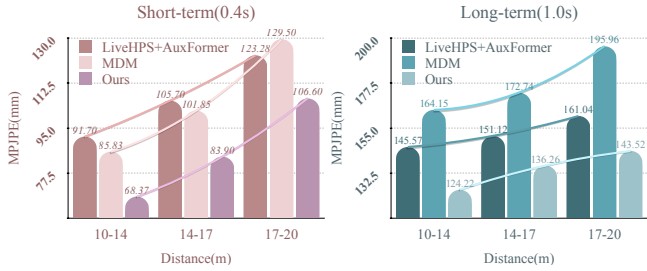

**Figure 4: Evaluation for the generalization capability on various distances on LIPD.**

*(2) Occlusion and noise.* To evaluate the robustness of our algorithm against occlusions and noise, we conduct simulations on the LIPD dataset. We introduce random noise by adding 30 noise points around the human point clouds and apply noise to 20%, 40%, and 80% of the frames, respectively. For occlusion simulation, we

randomly mask points within a cubic region measuring 0.4 meters on each side on the human body, affecting 20%, 40%, and 80% of the frames respectively. The results, as presented in Tab. 7 and Tab. 8, demonstrate that even under severe 80% noise or occlusion, the performance of our algorithm only slightly decreases, underscoring its robustness to noise and occlusions. *More visualization results on noise and occlusion cases are detailed in our supplementary.*

*(3) Real-world applications.* Fig. 1 shows that our method is practical for in-the-wild scenarios, capturing human motion in real-world scenarios day and night with real-time performance about 40 fps. This strongly demonstrates the feasibility and superiority of our method in real-life applications.

### 4.6 Diverse Motion Prediction Extension

Given the inherent spontaneity and unpredictability of human behavior, deterministic action predictions, which provide only a single outcome, are not optimal for scenarios requiring the anticipation of multiple possible outcomes for accurate decision-making. To address this, we expand our framework to diverse human motion predictions, covering up to four potential future motions. Our comparative analysis of deterministic and diverse motion predictions, detailed in Table 6, reveals that as the prediction time horizon increases, so does the uncertainty in human motion. This underscores the benefits of adopting diverse predictions, particularly for long-term forecasting. *Visualizations of these diverse motion prediction results can be found in our supplementary* materials.

### 5 CONCLUSION

We introduce the first single-LiDAR-based method for practical 3D human motion prediction. Building upon our effective structure-aware body feature descriptor, our approach adaptively maps the observed motion manifold to the future and models the spatial-temporal correlations of the human motion for further refinement. Additionally, we extend our method to support diverse predictions, accommodating multiple potential future motions for improved decision-making in real-world applications. Extensive experiments validate our method's superior performance and generalization capabilities in real-world human motion prediction scenarios.

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
