# OpenReview forum: "Towards Practical Human Motion Prediction with LiDAR Point Clouds"
_acmmm.org/ACMMM/2024/Conference — MM2024 Oral_

### Official Review · Reviewer_va2C · 2024-05-22

**Rating:** 4
**Confidence:** 3

**Summary:**

This paper presents a novel one stage point-cloud-based human motion prediction pipeline, which tries to address the problem of accumulation error observed in two stage pipelines. The experimental results seem to show the superiority of the proposed method.

**Strengths:**

+ The motivation of this paper is clear. The two-stage methods may suffer the problem of accumulation error, so this paper proposes a one stage method.
+ The presentation of this paper is good. The writing and the presentation of figures are clear and food.
+ It seems that the proposed pipeline achieves great performance and can be embedded into a real time equipment.

The reviewer thinks it can be a good paper, and expects the authors to address the concerns. Although the current rating is not high, the reviewer has a strong willingness to give a WA rating after the authors address their concerns.

**Limitations:**

- The reviewer wants to know why previous methods follow the two-stage solution rather than the proposed one stage solution. It may be better to add the discussions about the challenges of one stage solution.
- What is the metric for evaluating the performance? The reviewer does not find the explanation for the metric used in the tables, and does not know how the data in the tables are computed.
- As section 3.6 mentioned, the proposed pipeline can generate multiple predictions and uses WTA in the training stage. How to select the final prediction during the inference? How can the number of queries influences the final performance? The reviewer thinks these missing information is important.

To be honest, the presentation makes me feel that this is a good paper that should be published in the ACM MM conference. But some important information is missing, so the reviewer currently provides a BA rating. The reviewer really expects the missing information can be added, and will change the rating to WA.

**Suitability:**

3

---

### Official Review · Reviewer_c5Vm · 2024-05-25

**Rating:** 4
**Confidence:** 3

**Summary:**

This paper proposes a LiDAR-based model for 3D human motion prediction by fully utilizing the spatial-temporal information of point cloud frames. It achieves SoTA performances on two long-term motion prediction datasets.

**Strengths:**

1. The paper proposes a novel LiDAR-based framework to predict human motion.
2. It exploits the potential of spatial-temporal information of point cloud frames to enhance the sequential prediction.

**Limitations:**

1. Some parts of this paper are trivial: (1) the teaser image (Figure 1) provides abundant information to the audience and is easy to be confused. Additionally, your proposed work is a multi-stage framework as well (prediction + refinement), so it is better to clearly demonstrate the difference with the previous two-stage method at the left-upper corner sub-image; (2) the overall summary of "Section 4 EXPERIMENTS" (line 550-563) is useless.

2. The model designs for the spatial-temporal correlation (Adaptive Motion Latent Mapping and Spatial-temporal Correlations Refinement) are not clearly illustrated in Figure 2, although they are explained in detail in the manuscript.

3. The ablation study is conducted only on the LIPD dataset, not on the LiDARHuman26M dataset.

**Suitability:**

3

---

### Official Review · Reviewer_oABa · 2024-05-28

**Rating:** 5
**Confidence:** 2

**Summary:**

This paper proposed a single-LiDAR-based 3D motion prediction method, namely LiDAR-HMP, to directly predict the future 3D human pose from the raw LiDAR point cloud. To achieve this, a structure-aware body feature descriptor is proposed to encode the human body structure's global and local representation by leveraging a point encoder and human point cloud parser, respectively. After that, an adaptive motion latent mapping module is designed to regress coarse 3D human joints by querying the features from the past frame descriptors. Furthermore, a spatial-temporal transformer is proposed to refine the predictions using spatiotemporal contextual correlations.  Extensive experiments demonstrate the superiority of the proposed method in comparison with state-of-the-art on two benchmark datasets.

**Strengths:**

1. The paper is well-written and easy to follow.
2. The proposed modules in the overall pipeline sound reasonable and the motivation is clear.
3. Comprehensive experiments have been performed and demonstrate the superiority of the proposed method.

**Limitations:**

I do not find obvious technical weaknesses in this paper, but only have a few questions:

1. As claimed in the paper, the proposed method is the first LiDAR-based method for 3D human motion prediction from raw LiDAR point clouds directly. Why are there few works to do so? What are the challenges behind this?

2. It seems the proposed method can predict the 24 joint points of the 3D human pose from the LiDAR point clouds. So how to get the ground truth 3D joints $J_t$ for calculating loss?

3. As mentioned in Sec. 4.2, the input point cloud $P_i$ of each frame is set to be a fixed number, i.e., $N=256$, by using the farthest point sample algorithm. How the number of points in each frame can impact the performance of the 3D pose estimation?

4. I am confused by the setting of the $K$ in the structure-aware body feature descriptor, which is the number of bins corresponding to different body part segments. So is it determined or predefined by the human parsing model? In addition, in Sec. 4.2, implementation details, $K$ is set to 9, but in Sec. 3.6, for diverse motion prediction extension, $K=4$. Why is that?

5. Furthermore, seems the $K$ also controls the size of the final point cloud predictions $\hat{P}_t^{\prime} \in \mathbb{R}^{K\times32\times3} $, does it mean for each bin, the prediction head will output 32 3D points? BTW, seems the notation $^{\prime}$ is a typo.

6. Could authors elaborate more on the limitations of the proposed LiDAR-HMP and promising directions to enhance the performance in terms of, for example, scalability, flexibility, or applicability to other datasets or tasks?

7. Some typos, for example, in Sec. 4.6, line 910, “Table 6” should be “Tab. 6“.

**Suitability:**

3

---

### Official Review · Reviewer_wTko · 2024-06-05

**Rating:** 5
**Confidence:** 3

**Summary:**

Existing human motion prediction methods focus on forecasting future human movements based on past ground truth observations captured through multi-view cameras or dense IMUs, and use specialized network structures to address the spatial and temporal dependencies in structured skeleton sequences. This is often impractical for real scenarios where only raw visual sensor data is available. Some methods may experience performance degradation due to cumulative errors. Moreover, simplifying raw visual data into sparse key point representations significantly reduces information density, leading to the loss of fine-grained features. This paper proposes a single LiDAR-based 3D human motion prediction method that takes raw LiDAR point clouds as input and directly predicts future 3D human poses. The method consists of three parts: Structure-aware Body Feature Descriptor, Adaptive Motion Latent Mapping, and Spatial-Temporal Correlations Refinement. The first part addresses how to extract useful features related to human body structure from unordered LiDAR point clouds. The second part addresses how to predict future human motion based on historical observations. The third part addresses how to accurately simulate the spatial-temporal correlations between human joints. Finally, experiments were conducted on two public LiDAR datasets, proving the superior performance of LiDAR-HMP in both short-term and long-term human motion prediction.

**Strengths:**

1. LiDAR-HMP directly utilizes raw LiDAR point cloud data for human motion prediction, eliminating the need for preliminary pose estimation steps, simplifying the prediction process, and reducing the accumulation of errors.
2. The Structure-aware Body Feature Descriptor can deeply comprehend the human body structure and extract expressive features, which helps in more accurately capturing human dynamics. Adaptive Motion Latent Mapping allows the method to dynamically extract information from historical data and adaptively predict future movements, enhancing the flexibility and accuracy of the predictions. The Spatial-Temporal Correlations Refinement module takes into account the spatial and temporal correlations between human joints, further improving the precision of the forecasts.
3. Experimental results show that LiDAR-HMP maintains effectiveness under varying lighting conditions and real-time requirements, demonstrating good robustness. Moreover, LiDAR-HMP has expanded support for diverse motion predictions, capable of providing multiple potential future motion forecasts, which is extremely useful for application scenarios that require consideration of various possible outcomes.

**Limitations:**

1.	This article directly processes LiDAR point cloud data. When occluding the point cloud data, is the occlusion position random or targeted? Has the impact of occluding key body parts on the prediction results been considered?
2.	This paper directly processes LiDAR point cloud data, which can generate a substantial volume of data, requiring considerable computational power and storage space for processing and analysis.
3.	The paper mentions the deployment of the proposed scheme in real scenarios, but it does not provide a comparative analysis of actual performance metrics such as test time consumption and model file size, as seen in reference [28]. Complex models require significant computational resources, which may limit their application on devices with limited resources.

**Suitability:**

3

---

### Meta-Review · Area_Chair_PAXd · 2024-07-01

**Recommendation:** Accept (Oral)
**Confidence:** 4

**Metareview:**

Reviewers' concerns are well addressed in the rebuttal. All reviewers agree to accept the paper. The AC recommends acceptance.